



# 1 Educational and artistic fun teaching tools for science outreach.

Marina Locritani[1,3], Silvia Merlino[2], Sara Garvani[1,3], Francesca Di Laura[4].
1 - Istituto Nazionale di Geofisica e Vulcanologia, Roma 2
2 - Istituto di Scienze Marine, Consiglio Nazionale delle Ricerche, Pozzuolo di Lerici
3 - Historical Oceanography Society c/o Istituto Nazionale di Geofisica e Vulcanologia, Roma 2
4 - Istituto Nazionale di Geofisica e Vulcanologia, Amministrazione Centrale, Roma
*Correspondence to*: Marina Locritani (marina.locritani@ingv.it)
**Abstract.** The aim of scientific dissemination is to spread interest and knowledge of scientific issues by trying to reach
people of all ages and social backgrounds. Simplifying, without trivializing, scientific concepts and making them
attractive to the general public is therefore essential to achieve the previous objectives. For this purpose, it can be useful
for scientists to work in close collaboration with artists, implementing new tools that can positively influence the
emotional sphere and capture the attention of the people involved. Playful educational activity and visual language play
a key role in this process, to convey interest and facilitate learning. An example of this approach are the educational
laboratories structured as group games, in which great importance is given both to practical activities and to the
transmission of concepts through their visualization in the form of images. Over the last eight years, the Istituto Nazionale
di Geofisica e Vulcanologia (National Institute of Geophysics and Volcanology), the Institute of Marine Sciences of the
National Research Council and Historical Oceanography Society have collaborated in the organization of science
dissemination events involving students from schools of different levels in educational experiences based on games,
characterized by an essentially visual approach to the concepts presented. In this work, we would like to give a brief
overview of these didactic tools, retracing the choices made while ideating them, thanks mainly to the close collaboration
with some artists and illustrators.

## 24 1 Introduction

**25 The role of visual language in the translation and synthesis of scientific content**

Inspiring young people to take part in the discovery and delivery of science, technology, engineering, and mathematics
(STEM) has been proven to contribute significantly not only to their well-being, but also to their future human
development (Bertram and Pascal, 2016; Morgan et al., 2016; Friedman-Krauss et al., 2018). Especially primary and
secondary education were considered significant periods for the development of students' interest in science and
technology (Maltese et al., 2014). In recent years, with the advance of the digital age and the use of technological tools
(smartphones, tablets, etc.), now pervasive especially among the new generations, great importance has been given to the
development of strategies to promote their use in schools: an attempt has been made to convert them into useful means to
promote information and knowledge, especially those related to STEM, and so to overcome the difficulties observed in
the teaching-learning process (Souza et al., 2018).
However, the "physicality" of the experience is still important in our opinion, and, even more so (given the increasing
migration of interest and experience towards the virtual world), it is necessary to invest resources to create stimuli that
involve students in real activities. The search for new strategies to communicate to students the importance of STEM is
a fundamental step to improve their learning experience and to provide alternative teaching practices to teachers. On this
basis, we asked ourselves the question: "is it possible to enhance the learning experiences of STEM disciplines for students



(6-14 years old), using a visual approach that serves to stimulate interest in the proposed topics? Visual language has
always been the most comprehensible for everyone; for this reason, a lot of modern (Cavallo F. and Favilli E., 2016) and
ancient (Merian M.S., 1705) authors use images as a tool to convey scientific notions and findings. Since its origins,
science has placed images at the centre of its communication processes: drawings, diagrams and then photographs,
satellite images and films. Studies on the complexity of nature, the human figure and the technological innovation of
famous people, such as Leonardo Da Vinci, highlight the scientific and, at the same, time artistic value of drawing. With
the transition from the empirical to the experimental approach, images retain their value. Today researchers use images
for the interpretation of collected data: in this way maps and diagrams become indispensable for the scientific process. In
parallel, some of these images (such as those of Hooke R., 1665) for their uniqueness can be considered works of art.
Moreover, in recent years, two strongly conditioning factors have taken over in modern communication: the speed and
amount of information we are exposed to. In this fast-paced world, not only for adults but also for of childhood, it is clear
how much more effective a message conveyed by the image is than the text and how much faster its learning is. By now,
in fact, it is universally recognized the effectiveness of images not only in the communicative-advertising field, where
they have always been widely used and even more so in modern society, constantly exposed to visual information in the
form of video or images, but also in the world of scientific communication.
For sixteen years now, on an annual basis, the Science Technology and Observa Science in Society Monitor
(www.observa.it) has been monitoring the development of so-called "scientific literacy", i.e. the level of scientific
knowledge of citizens. Much less studied is the so-called visual scientific literacy. In 2016, an empirical survey (Bucchi
and Saracino, 2016) was conducted on this topic on a representative sample of the Italian population: the interviewees
were offered three classic images related to science and technology, in regards to a series of questions of scientific
competence. With regard to the matter under investigation it emerges that the level of scientific literacy decreases with
increasing age and increases with increasing education. From the point of view of the effectiveness of the visual approach,
the results show that 80% of respondents were able to recognize the images correctly, compared to 60% who were able
to answer the equivalent questions. Moreover, the images, unlike the questions, aroused emotional reactions such as
"curiosity", but also "beauty" and "fear". This highlights the enormous potential of the visual component in scientific
communication, whose characteristics, dynamics and means of dissemination must be fully understood in order to obtain
even more significant results.
The cooperation of such different worlds, such as art and science, and the exchange between the two points of view,
allows the disseminator to develop a new approach to scientific issues, generating a common language that transforms
complex concepts into visual messages that can be understood by all. In particular, in order to capture the attention of
children, the goal is to create games that convey, through images and oral explanation, in an emotional and non-rational
way, information related not only to scientific knowledge but also to the learning of virtuous behaviour, which will allow
new generations to become adults more aware of the environment in which they live, how to use it and how to preserve
it.
This document is organized as follows: Section 2 addresses the problem of teaching and dissemination of scientific
culture, and presents a description of the methodological approach we propose; Section 3 describes the characteristics of
the different edutainment tools implemented, with particular attention to the different graphic choices adopted according
to the aim to be achieved; Section 4 describes, as a case study, a "work-related learning internship" that we carried out
using one of our educational tools, and reports the result of a questionnaire that we submitted to the students at the end of
the activities; section 5 summarizes the main conclusions on what has been done in recent years, and the future prospects.



## 2 Science and education today

### 2.1 The importance to teach science in today's scenario

Recommendation 2006/962/EC makes explicit the EC support to each Member State for the development of education and training strategies that follow a specific and harmonised path offering everyone the opportunity to develop their basic competences in the form of knowledge, skills, abilities and attitudes, while engaging in active and democratic participation in society (especially in increasingly diverse societies). STEM competence is the third of the 8 key competences that this recommendation has identified as fundamental for each individual in a knowledge-based society, because "*Science, being one of the most remarkable achievement of human culture" and must "…be shared with all, especially when extreme specialization of scientific disciplines and complexity of their results seem to hopelessly increase the gap between science and the average person*" (Wilgenbus and Lena 2011). Unfortunately, the special edition of the 2014 Eurobarometer on public perception of science, research and innovation (Special Eurobarometer 419 Report, 2014), indicates, especially for the Italian population, a low interest in science and therefore a lack of confidence in the potential of research. In particular, although the results on the one hand are rather encouraging and show Europeans (79%) and, to a lesser extent, Italians (69%), very interested and confident in new scientific discoveries and technological developments, however there is a part of respondents who do not feel as well informed (52% Italians, 50% Europeans). A low percentage (31%) of Italians and even fewer Europeans (22%) believe that science can solve any kind of problem; a part of the population (52% Italians, 58% Europeans) would like researchers to be more involved in the transmission of scientific discoveries and new technological developments. Another important fact is that 75% of respondents believe that science prepares future generations to act as aware citizens, and many citizens (65% Italians, 66% Europeans) think that the government should stimulate more young people's interest in science to a greater extent. Finally: a high percentage of Italians (71%) and Europeans (75%) agree that if women were more represented in positions of power in research institutions, research would be conducted in a better way.

The Eurobarometer, therefore, confirms a situation of "disconnection" between civil society and science, which, being one of the most remarkable expressions of the realization of human culture, should, instead, be shared with everyone, especially when the high level of complexity of the results could further increase this gap (Wilgenbus and Léna, 2011).

How to act, therefore, effectively? Fostering education, awareness and dissemination through simple and attractive channels capable of reaching every level of society and different age groups, with particular attention to the younger generations. As already said in the previous paragraph, currently it emerges that the level of scientific literacy decreases with increasing age and increases with increasing education (Bucchi and Saracino, 2016); therefore, the importance of planning a process of scientific literacy from the early school years is evident. Unfortunately, in several countries, such as the United Kingdom, STEM topics do not appear on the timetables of pupils of primary or lower secondary school (Bianchi and Chippindall, 2018). This gap could be filled by giving schools the opportunity to be involved in extracurricular programmes, promoted by researchers or educational trainers with scientific expertise, always taking into account the prerogatives of children of that age. Kids are an important vector for messages aimed at social change: it is therefore unthinkable not to take into account attitudes and decisions that will inevitably affect the environment and civilization of the future (Hartley et al., 2015).

### 2.2 How to foster STEM education?

### 2.2.1 Learning through play



By playing, we learn to learn. This concept was introduced in the 1970s by Gregory Bateson (Bateson, 1970) and is
generally used to indicate the acquisition of a learning method that produces a change in the person. The game can
therefore be thought of as one of the simple and attractive channels mentioned in the previous paragraph. It is necessary
to underline the importance of the relational factor of learning, which implies the interpretation of the experience lived
through patterns learned in contexts of communication and interaction with others (Vygotskij, 1933).
In this context, playing with an expert, whether an adult or a peer, takes on great educational importance, and is the very
driver of the child's development. Moreover, through play, we learn that we can give different interpretations of the world
around us (Braglia, 2011), and this helps children to grow up critical and more aware of the problems they will have to
face. For this reason, it is important to introduce, among the school teaching methods, also the involvement of students
in educational activities through a playful-scientific approach.
These activities must not replace the classroom lessons, books, and tests (Shapiro et al., 2014), but have to provide another
parallel and sporadic learning strategy: a way to provide the differentiated learning experiences that students require to
find their inner motivation and fulfil their potential. In this framework, it is important to include visual and tactile stimuli,
to encourage and enhance the ability to observe, pay attention and memorize concepts (Renninger and Su, 2012).
Moreover, it is important to focus on the visual aspect of the proposed tools, as emotions deeply affect our cognitive
aspects: a welfare state has positive effects on learning ability, memory, creativity (Ellis et al. 1984). In fact, several
studies have underlined how the human capacity to understand is based not only on the faculty of reasoning, i.e. logic,
but also (and above all) on emotional mechanisms (Kahneman, 2012). Life experiences create somatic markers related to
emotions that guide us in decision making for successive events (Damasio, 1994). Affective neuroscience - that studies
brain emotions through non-invasive techniques of "imaging" - has shown that positive emotional states are developed
(such as optimism and joy) when the amygdala and the right prefrontal lobe raise their activity levels (Davidson 2002).
Similarly, studies on the brain biochemistry locate in the forebrain most of the neuropeptides and neurotransmitters
activities as well as the receptors responsible for the physiological sensations of well-being (Pert, 1999). Daniel Goleman
(Goleman, 1996) clearly defines "emotional intelligence" and social-emotional learning as a balanced mix of motivation,
empathy, logic and self-control. This should be taken into account in order to develop much more effective and
appropriate methods of science communication/education, depending on the interlocutors (age, nationality, previous
knowledge, gender, etc.). By way of example, we can consider that of the stereotypes used, in too many cases, by teachers
and/or syllables, to represent scientists and generally those working in the scientific field, which inadvertently discourage
potential female STEM students (Petkova K. & P. Boyadjieva 1994, Newton and Newton 1992).

**2.2.2 The use of visual arts as a support for science education**

If, as we have seen, the game can be useful from a relational point of view, especially among children and teenagers, it is
also necessary to establish a language that is common to all involved, if we want our teaching methods and tools to be as
sharable as possible. A common and comprehensible language for everyone can be found in visual arts, but translation of
concepts in images which everyone can understand and memorize, also under an emotional point of view, can be a quite
challenging task, especially when the objective is suggesting changes in mentality or building a learning process. In fact,
it is difficult to assess and predict some key aspects related to human vision. For example, how visual information is
perceived is affected by the specific training of each one and contingent upon historical, geographical and cultural
circumstance. Moreover, the personal experience allows to emotionally differently interpreting the images (Geymonat,



2011). In any case, graphic art has become effective in many different situations, and this has made it the preferred
language of the new generation.
For this reason, we decided to strongly characterize our didactic tools focusing not only on the contents, but also on the
graphic aspect, trying to express as much as possible the concepts and themes proposed through visual approaches, and
making them pleasant and suitable, from a graphic point of view, to an audience of children/youngsters. The choice of
drawings, layout and colours followed these considerations, leading to the adoption of different graphic/artistic
techniques, depending on the target audience and the mean of communication. In the following paragraphs, we would
like to present 4 of the products currently made, namely three educational games and a graphic questionnaire.

**3 Inside the tools: educational purpose and visual approach**

The four activities that we present were developed during 9 years, from 2011 to 2020. These are:

- 3 Customized games to achieve specific learning goals (for Edutainment Activities);
- 1 VISUAL QUESTIONNAIRE (for Evaluation of children's Science Perception).

The first Edutainment Activity (OCTOPUS GAME) is specially designed for reaching children and students of Primary
Schools. The second one (INGV-MEMORY GAME) is designed for students of low secondary school (middle school),
and the third proposed game, MAREOPOLI, for the content presented, proves to be more suitable for student of the high
school. This division by age is to be considered strict if the game is played by the students at home alone or with their
parents, but it is more elastic if the game is played directly by a skilled facilitator (researcher or teacher) who can modulate
the difficulty of the questions and answers but not changing the content. The last tool, the graphic questionnaire, is
intended for very young children: it is based on the game of paper puppets with interchangeable clothes and furnishings,
and allows the player to freely create and invent its own "scientific character" and the work/creative environment in which
it is placed.
The starting point, for each designed tool, has been the definition of the educational goal, which was different in the four
cases: in the OCTOPUS GAME the objective was the dissemination of knowledge regarding marine environmental
protection, and some basic notions of marine biology and ecology; with INGV- MEMORY we wanted to give relevance
to the three main lines of research developed by the INGV: Earthquakes, Volcanoes and Environment; in the
MAREOPOLI the objective was, instead, to enhance the Historical Oceanographic Heritage, extrapolating from ancient
texts the evolution of the tide theory and comparing it with the contemporary scientific explanation; finally, with the
VISUAL QUESTIONNAIRE, we would like to investigate the perception of science and scientists in very young children,
as primary school students or pre-schoolers. In this case, the use of a visual approach is preferred to a standard
questionnaire with closed-ended questions.
Once we defined the educational goal, we chose the type of game to be inspired to develop the structure of the didactic
tool. For the 3 board games, we have chosen "The Goose Game" for the first one, "MEMORY ® of Ravensburger" for
the second one and "MONOPOLY ® of Hasbro" for the last one. In all three cases, we added, to the standard structure
of the game, specific scientific information and questions, and planned it so that it could be played by two teams.
We have considered important to divide the players into teams, in order to establish a competitive dynamic among
children, but without mortifying a single player in case he can't answer the questions correctly. The games have been




designed and built in two formats (laboratory mode and game kit), in order to consent to be used, by students, in two
ways: under the guidance of trainers (researchers, teachers or suitably educated young people during peer education
approaches) during the didactic workshops, or independently (or under the guidance of their parents) at home. For the
first purpose (laboratory mode), therefore, boards and related play-cards have been realized in a very large format, suitable
to be placed on the floor, in order to give the opportunity to children to feel more involved during the activity and better
visualize the drawings without losing attention. At the same time, also "table-top" formats (game kit) have been prepared,
to be distributed as gifts to the participants of the workshop, inviting them to disclose, in turn, the information acquired
during the activity to friends and relatives. Moreover, for MAREOPOLI a dissemination book to deepen the information
was made.

The realization of the materials was made thanks to a close collaboration with the graphic designers, who have translated
into images the researchers' ideas.
For their part, researchers initially had to simplify (but without trivializing) the concepts and devise understandable, but
at the same time engaging and informative, questions. The graphic designers tried to visually imagine the questions and
concepts, proposing solutions and graphic styles suitable for the age group for which the game was intended and the topics
addressed. This part of the design was very interesting for both parties involved (researchers and designers), but not at all
simple (Figure 1).

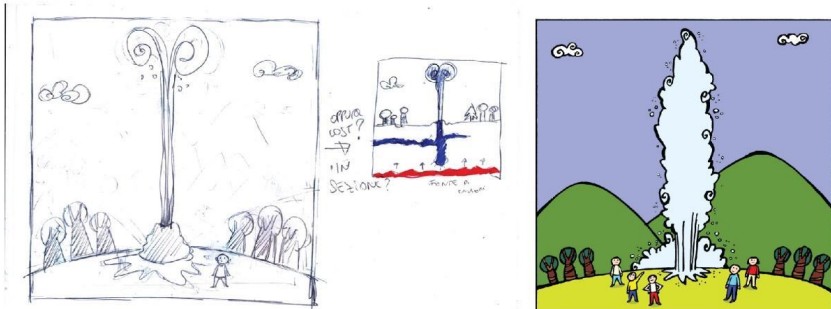

**Figure 1. MEMORY graphic draft for playing card geyser. The artist proposed two different solutions to the researcher, and**
**together decided the final illustration (Illustration made by Consuelo Zatta).**

In fact, a scientific concept or argument can already be intrinsically not easy and immediate to understand, and trying to
express it through a static image is not immediate, but it is precisely here that the expressive capacity and universality of
art comes into play, which often manages to reach where words do not arrive. In some cases, the images only had to be
the background to the scientific theme; but even in this case, the choice of how to represent it and make it attractive
stimulated the minds of researchers and artists involved in this task. In other cases, (the most difficult ones) the graphic
itself had to indicate and help to understand the concept or the application submitted: the teamwork between the two
different skills (scientist and artist) was, so, particularly important and essential, made of continuous adjustments and
corrections, until obtaining a final product that could meet the graphic and conceptual requirements.

**3.1 Specific description of didactic tools**
**3.1.1- OCTOPUS GAME**




The OCTOPUS GAME, based on the board traditional Game of the Goose, consists of a big board of 2 x 2 meters,
containing a round track with 20 spaces numbered counter clockwise. Three colours have been used for the different
spaces, associated to three different topics. Some spaces have symbols that correspond to specific indications how to
move inside the board. Each space has a different question (Figure 2), indicated in a separate game-kit. When choosing
the graphic layout, drawings, accessories (dice, placeholders) and colours for this game, the preferences of children of the
age in question have been taken into account. The spaces have 3 possible colours: blue, green, and yellow, each one
corresponding to a specific topic, intuitively linked to the respective colour: water column, life in the sea and coast and
seabed. The best-known characteristics of some animals have been used to indicate the special spaces such as, for
example, the shrimp, which makes the player jump in two spaces, or the jellyfish, which (for pain) makes the player stop
for a round.

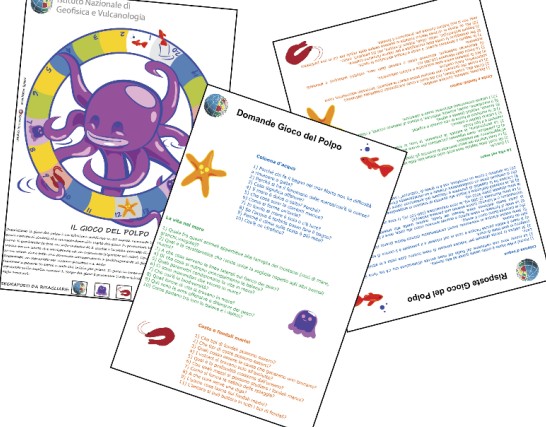


**Figure 2. The kit-game of OCTOPUS GAME. One paper includes the board and the rules of the game. Another paper**
**includes the questions and the correspondent answers about marine water column, marine life and seabed and coast.**
**(Illustrations made by Matteo Sgherri).**

The educational laboratory is organized as a competition between two teams, to be carried out under the supervision of
an expert, who holds the ranks of the game, asks the questions and guides the teams towards the correct answer (Figure
3). It takes place in about one hour. The children, using large dice, extract numbers to advance the placeholder boats
(made of coloured sheets of paper) on the board. Players take turns to roll the dice and move their piece forward by the
sum of the two dice. Every time a placeholder stops on a space of a special colour, a question corresponding to the
associated topic is asked to the group on duty. During this path, children have fun and, thanks to the spirit of competition
that is created between the two groups, try to actively answer the questions asked. At the end of the educational laboratory,
play kits are distributed (Figure 2) that allow boys and girls to play at home with the help of their parents.





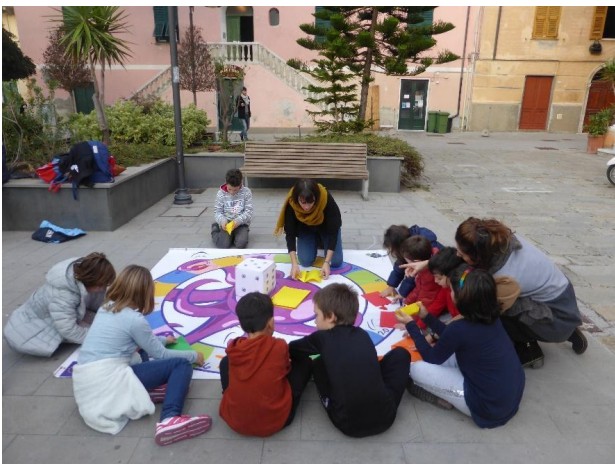

**Figure 3. A group of students play at OCTIPUS GAME with a INGV researcher ("conductor" of the game), during an outreach**
**event in the city of Monterosso (La Spezia).**
What happens if respondents don't understand a questions? In this case, the conductor of the game starts asking other
questions related to the first but simpler one, trying to refer to practical examples, easily referable to everyday life. In this
way the group always, or in most cases, manages to give the right answer.
The role of the "conductor" during the educational laboratory is therefore important, but it does not necessarily have to
be a researcher or a teacher. In fact, the game has also been experimented during peer education courses, with students
older than the involved children. In this case, the experience has a double objective, because the learning by teaching
mode has proved to be an efficient method to stimulate learning, as it leads to an increase in self-esteem and self-
confidence, which in turn conveys a greater retention of the concepts acquired during the experience. Since the topics
dealt with in this game are quite simple, it lends itself well to be used in peer education mode, since it does not put students
in difficulty and, instead, leads to strengthen their knowledge in the field and also to discover new things. The playful
aspect of the experience is stimulating not only for the children involved in the workshop, but also for the peer educator
himself.
**3.1.2 INGV-MEMORY**
The INGV MEMORY GAME is a board game based on the classic Memory ® of Ravensburger and have the same game
rules. It has been especially designed to help to improve concentration and train visual memory by turning over pairs of
matching cards; at the same time children must also associate images with some basic concepts on Vulcanology,
Geophysics and Environment, with particular attention to natural hazards: Volcanoes, Earthquakes and Tsunamis. A
game-kit is available (Figure 4).



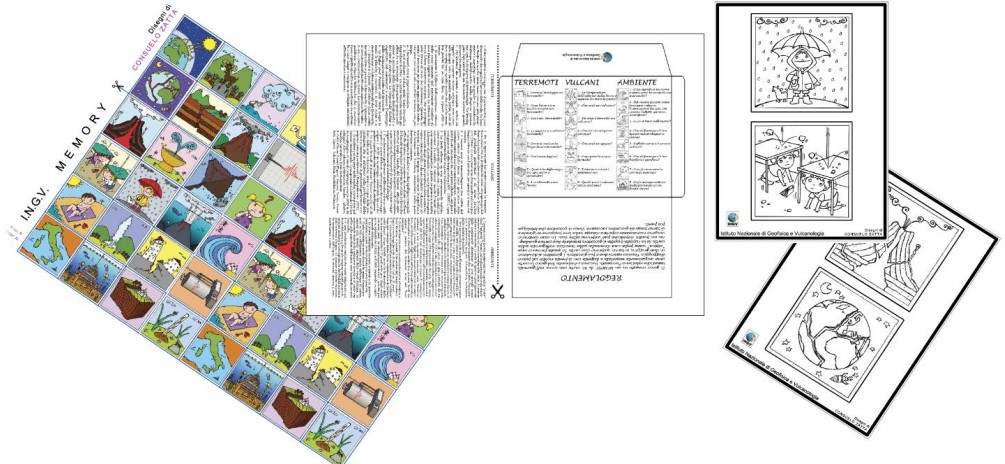

**Figure 4. INGV-MEMORY kit game. The kit includes 48 cut-out cards, the correspondent questions and answers and the**
**colouring sheets (Illustration made by Consuelo Zatta).**

The game consists of 48 cards (size 40 cm x 40 cm each) with icons depicting the covered topics: Earthquakes, Volcanoes
and Environment. Also in this game, the dynamics during the educational laboratory mode is that of a team game: the
cards are initially shuffled and laid out covered on the floor. The players are divided into two groups and, in turn, reveal
two cards. If these forms a "pair" of matching cards, the researcher/conductor asks a question about the theme depicted
on the paper. If the answer of the team is correct, the cards are cashed by the player on duty, who can uncover two more;
otherwise, they're placed back in their original position on the floor, and the turn move on to the next player. The player
who can discover the more pairs wins.
The game is preferably aimed at middle schools (Figure 5). As for the OCTOPUS GAME, also for the INGV-MEMORY
GAME the educational laboratory takes place in one hour during which the students will be able to test their mnemonic
skills and scientific knowledge.
The graphic of this game is more polished than the OCTOPUS GAME. The figures are better defined, even if the colours
and the typology are always referable to the cartoon style, or however typical of the illustrations for children.
In this case, however, what is represented in the different boxes has to do with the proposed topic, and sometimes it is
also useful to understand and interpret the asked questions. The graphic and artistic challenge of interpreting the scientific
concepts was therefore more complex, compared to what was done in the OCTOPUS GAME.



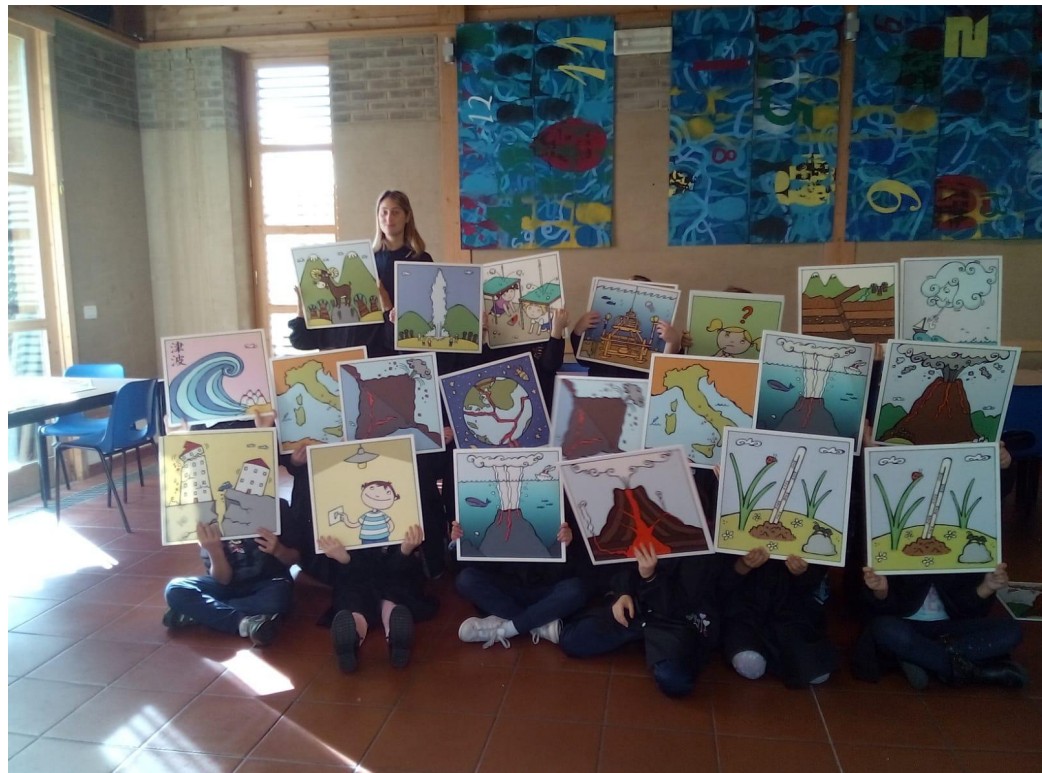

**Figure 5. The educational laboratory INGV-MEMORY GAME at the "*Istituto comprensivo di Vezzano Ligure – ISA11*".**

In this case, too, at the end of the educational laboratory a game kit is released to each student (Figure 4) that allows to play at home.

**3.1.3 MAREOPOLI**

MAREOPOLI (Figure 6) is a reinterpretation of the famous board game MONOPOLY ® of Hasbro. It was created with the aim of spreading the knowledge of the Historical Oceanography and the scientific path of tidal theories, from the Greek period until the end of the eighteenth century. Many scholars, in fact, have tried to understand and interpret this phenomenon. Among the most ancient we can mention Aristotle and Eratosthenes, but also seventeenth-century figures such as Galileo Galilei, up to the physicists who have formulated modern theory like Newton and Laplace. The importance of the historical basis of our knowledge is an issue that is very close to our hearts, and which is often not sufficiently highlighted.

The INGV houses a collection of historical oceanographic books ranging from 1494 to 1799, some of them of inestimable historical and artistic value, with handmade drawings and xilographies. Part of the graphic material created for this game (the curiosity cards), therefore, has been designed so that it can be extrapolated from the game and used as a comics book for adults, which reconstructs the entire history of the evolution of tidal theory from Aristotle to Laplace (Figure 7). This aim of public awareness of science and historical knowledge is combined with the educational aim of providing scientific



information on the history and scientific theory of tides, but also on transversal but tide related issues, such as: renewable
energy, biodiversity, protection of the planet.
The board created for this game recalls the graphics of the MONOPOLY (Figure 6). The board format for playing it in
groups during edutainment laboratories measures 2 x 2 meters. The board is made up of 36 spaces: the space **GO**, 16
spaces **city,** 18 spaces **curiosities** and 18 spaces **unforeseen.** To each space **curiosities** correspond a **curiosities** card
(Figure 7) with notions of historical and general knowledge on the tides, while for each **unforeseen** space correspond an
**unforeseen** card, with scientific questions on tides; to each **city** space correspond 16 **city** cards describing the tidal
phenomena typical for those real cities in the world.

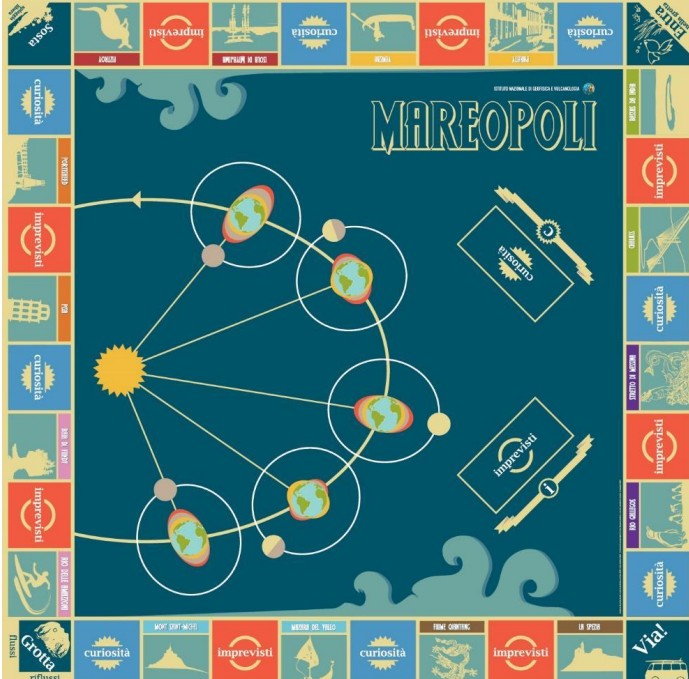


**Figure 6. MAREOPOLI GAME. The board, the curiosity, unforeseen and city cards explain the tide formation and the**
**evolution of tide theory (Illustration made by Francesca di Laura from INGV Graphic Office).**

The game is aimed at students of Second Grade Secondary Schools but, as already said, also at adults, thanks to the fact
that the curiosity cards have been processed in such a way that, if extrapolated from the game, they can be used as a
comics book, a graphic-artistic tale of the history of the scientific tides theory evolution.



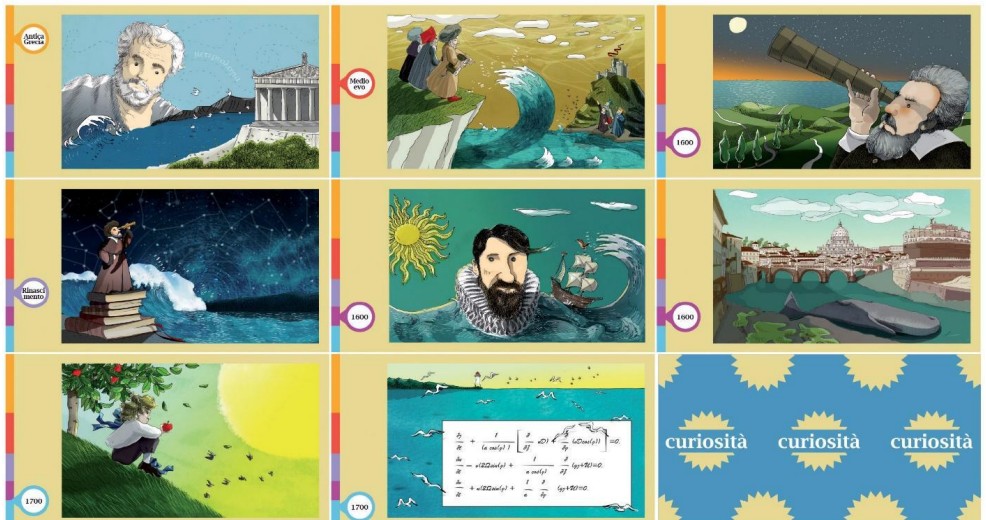

**Figure 7. The curiosity cards, if extrapolated from the game, can be used as a comics book, a graphic-artistic tale of the history of the scientific tides theory evolution (Illustrations made by Francesca di Laura from INGV Graphic Office).**

The twofold objective behind the planning of this tool led us to develop a particular graphic style as well. The graphic layout of MAREOPOLI appears less friendly than that of the two previous games: the drawings are more refined and made with a less comic book style, to be appreciated by adult users.

The game is therefore the result of the work and cooperation of scientists and illustrators, who have shared information, ideas and images to get the final product (Figure 8).

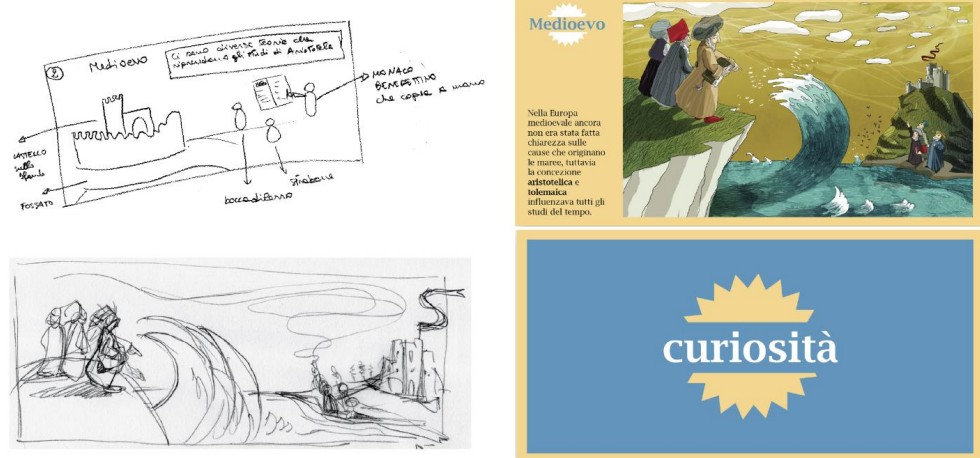

**Figure 8. Different stages of making a curiosity card. The first draft, at the top left, is that of the researcher who tries to convey to the illustrator his idea of graphics (the graphic storyboard). The others are the different proposals and evolutions made by the illustrator (Illustrations made by Francesca di Laura from INGV Graphic Office).**





The scoreboard itself looks very "sober" but elegant at the same time. The choice of colours was made also with end users in mind, so not very young children: the colours are not gaudy, and they have some taste tones a bit retro' that bring us mentally back in time, or at least emotionally approach different eras.

Also in this case there is a "board" game kit to give as a present, but, compared to the two previous games, it is supported by a dissemination book which deepens all the issues addressed in the game and the correlated fields.

### 3.1.4 VISUAL QUESTIONNAIRE

The VISUAL QUESTIONNAIRE has been elaborated in collaboration with Liguria Cluster of Maritime Technologies (DLTM), Centro Supporto Sperimentazione Navale - Italian Navy (CSSN), CNR-ISMAR, INGV and Steam Factory (private company) in order to assess Science Perception in school students of very low scholastic levels (primary or pre-scholastic age), and to understand whether participation in extracurricular dissemination events contributes in increasing interest in science subjects (Schibec, 2006). The project was the second step of a previous study that investigated Science Perception in older students with a write questionnaire (Locritani et al. 2015). In the specific case of 6-7 years old kids (that of course are too young to compile a questionnaire with closed ended questions), acquired stereotypical images were usually monitored trough the Draw-a-Scientist Test (DAST) (Figure 9). DAST is time-consuming and doesn't allow discerning in a quick way the child's response. In fact, DAST needs always to be coupled with an interview for posterior interpretation of drawings. We decided, so, to use an alternative approach that goes beyond DAST: an entirely Graphic Projective Questionnaire, a paper kit with pre-drawn characters, accessories and clothes inspired to mainstream cartoons aesthetics, that allow children to assemble stereotypical personages, as in a mix-match game (Figure 10).

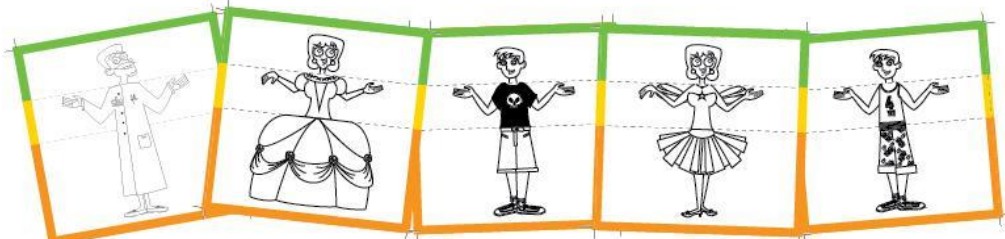

**Figure 9. Sketch of some of the available options for characterizing the clothing of the citizens of the Kingdom of Science. From left to right: old fashioned lab coat (science kingdom), princess dress (historical kingdom), feeling kingdom suit, fairy tale dress (fantasy kingdom), music kingdom dress. (Illustrations and graphic project made by Giacomo Guccinelli and Lucrezia Benvenuti, Locritani et al. 2015).**

A try-out phase has been necessary – with a group of 6 years-old children - in order to understand the background culture of this generation and language, pop culture, i.e. models and aesthetics references (TV, Web etc.), basic knowledge/perception of science (Saris and Gallhofer, 2007). Some direct questions allowed researchers to understand students' drawings, and in this phase DAST has been an essential initial tool for refining the VISUAL QUESTIONNAIRE. Each character is the result of the assemblage of 3 paper strips representing 5 different heads, 4 different body/arms, 4 different legs/feet and clothes and accessories (Figure 10). Each personage can be male, female, young, adult; moreover, a fifth neutral character halfway between the 4 previous characters as been added.



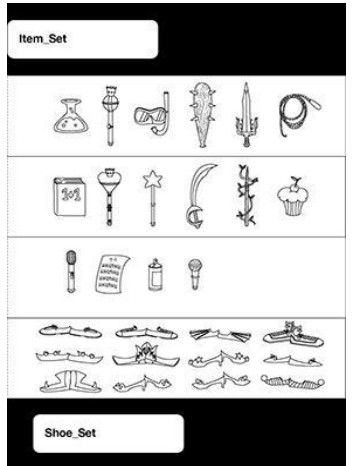

**Figure 10. Accessories of different characters (Illustrations and graphic project made by Giacomo Guccinelli and Lucrezia**
**Benvenuti, Locritani et al. 2015).**

The VISUAL QUESTIONNAIRE needed to be released from scientific context, so as not to affect the children response:
we've decided then to administer it through a story telling. The choice of the narrative makes questionnaire administration
more involving for children, which are free to use imagination, but at the same time are assisted in giving a clear response.
During the questionnaire administration, children are asked to represent themselves (an 'avatar'), the hero of the story
who visits five fantastic kingdoms (that cover different areas of the imagination and possible interests of children):
• Feelings
• Science
• History
• Music
• Fantasy
and the citizens living in these 5 kingdoms (Figure 11) for a total of 7 characters that will be photographed and analysed.
In this way, it will be possible to see how children identify themselves by seeking similarities amongst their own
representation, the hero and the citizens of the 5 kingdoms. A score chart will allow to have numerical results comparable
with those of the other questionnaire with closed ended questions, but this part is still a work in progress.

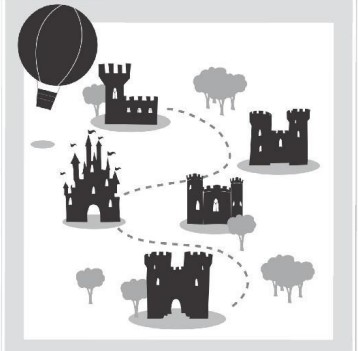




**Figure 11. Five kingdoms of Feelings, Science, History, Music and Fantasy (Illustration and graphic project made by Giacomo**
**Guccinelli and Lucrezia Benvenuti, Locritani et al. 2015).**

The VISUAL QUESTIONNAIRE is designed to give us information about on how children think and imagine science
and scientists, and is organized to be read and interpreted by "blocks":
• Block 1 Interest in Science subjects;
• Block 2 Projection of future personality;
• Block 3 Science Perception;
• Block 4 Perception about Scientists;
• Block 5 Interest in extracurricular activities.
In regards to the possible choices between clothes and accessories with which to dress and then interpret the figure of the
scientist, we have deliberately proposed those of the stereotype scientist, then white lab coats, ruffled hair, etc., but also
possible alternating choices, such as female figures, or modern lab coat, or scuba suit, etc. Today's scientist, even the one
proposed by the mass media, is different from the one of the past (typical "lab rat"), and this questionnaire will allow us
to understand if the social changes of the last years (and especially those related to female figures in many fields of work
once male prerogatives) have been received by children.
In this phase, the collaboration with unconventional educators (also expert draftsman) such as Guccinelli and Benvenuti
(Steam-Factory) was fundamental, in order to create the appropriate framework for the questionnaire, through the choice
of drawings in line with the expectations and preferences of the very young generations. These choices were in fact the
result of a process, prior to the creation of the VISUAL QUESTIONNAIRE, in which many children from kindergarten
and primary school were probed, in a discreet way, to understand their tastes and preferences in terms of graphics, art,
pop culture, etc.

**4 Case study - work related learning internship**

The previously described educational tools can be applied in a lot of contests, for example during outreach events,
scientific challenges, school activities, scientific festivals or High schools work related learning internship, using non-
formal methods as peer education and/or intergenerational learning with the support of unconventional educators.
Through a case study the efficacy of games and images to communicate science concepts will be evaluated. During a
"work-related learning" internships, regulated by a recent national Legislative Decree (DM 774 – September, 4th, 2019)
called PCTO (*Percorsi per le Competenze Trasversali e l'Orientamento*), 4 classes composed by 74 students have been
involved, by INGV researchers and GAD (*Gruppo Astronomia Digitale* – Digital Astronomy Group) experts, in 4
different types of activities: frontal lesson, practical activity, game and direct experience (planetary).
The PCTO aims is to provide general knowledge about some topics less faced by the school curricula: gravitational field,
astronomy and tides (from scientific and historical point of view). Frontal lessons about gravitational field and astronomy,
Planetary visit, an educational game about tides (MAREOPOLI) and a practical activity devoted to measure the
gravitational field, were performed.
After the PCTO the students compiled a questionnaire about the level of satisfaction. The questionnaire was elaborated
following the previous experience in this field (Locritani, et al. 2019).



Results show that the preferred topic (Figure 12) was astronomy (50%) followed by gravitational field (28,4%) and tides
(21,6). The most difficult topic was gravitational field (50%), followed by tides (29,7%) and astronomy (20,3%). Students
affirm an improving knowledge in gravitational field (41,9%), followed by tides (37,8%) and astronomy (20,3%).

**Level of satisfaction regarding topics**

**Figure 12. The histograms show the level of satisfaction regarding work related learning internship topics.**

The favourite activity (Figure 13) was the direct experience at Planetary (32%), the game MAREOPOLI (31%) and the
practical activity (30%).

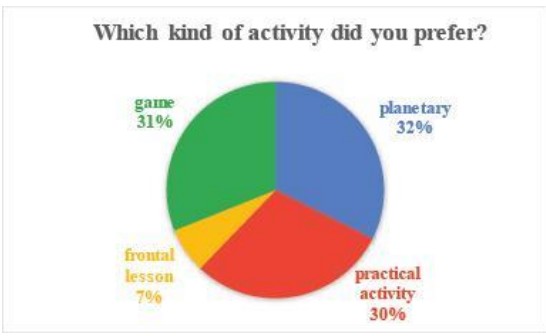

**Figure 13. The pie chart illustrates which kind of activity prefer the students.**

The questionnaire was designed to understand if for the student it was important the role played by the image in the
proposed activities during PCTO experience. In detail, three specific questions, about this theme, have been included in
the questionnaire (Figure 14 and 15): 1- Which is the role played by the images in the proposed activities? 2- Did a
particular image capture your attention? 3- Which one?
The results for the first question indicate an interest in image, in fact, student answers highlight that at 50% the images
capture the attention and at 43% the images facilitate learning.



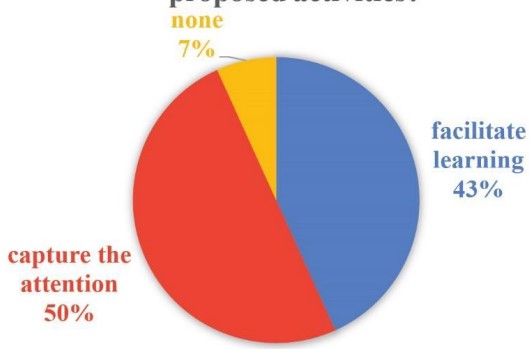


**Figure 14. The pie chart illustrates which role played the images for the students.**


The 59% of the students affirm that they do not remember with particular impression a specific image. Nevertheless, it is
interesting to note that each activity has influenced, by means of images, some students, and that the illustrations capture
the interest of the 45% of the students in different way. In details, about 7% of the students were impressed by images
included in the presentations, also if the presentation itself didn't arise the interest of the students (see above). This
indicates that images have a great power in capture attention, because students remember illustrations also if showed
during the "boring" presentations. The best result emerges for the images of the game, with 14% of interest (Figure 15).
In this case, students remember especially the historical characters, as Aristoteles or Newton, and the schemes used to
explain the tide formation. The historical characters illustrations have been created to focus attention on historical aspect
that, often, turn out to be very unattractive, especially for technical school students (as in this case). The schemes to
explain the formation of tides, instead, have been specially designed to simplify the concept (intrinsically hard) and make
them easier to understand, thus exploiting the potential offered by graphics compared to just written text.
We can assume that the questionnaire results, highlighting the importance of the image in keeping attention and in capture
the interest of the student, indicate that our objective has been reached.

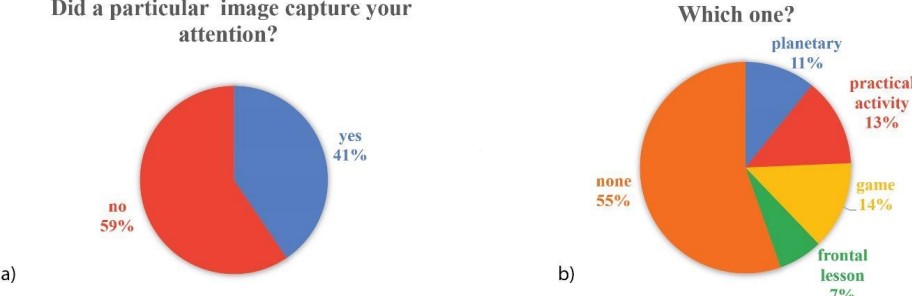


**Figure 15. The pie chart in the panel a) illustrates how much the students think they were captured by an image and the panel**
**b) shows which ones.**

**5 Conclusions**




The involvement of children and students in educational paths are useful methods to transmit scientific knowledge and
awareness about today's great environmental problem. In addition, they are also useful means of reaching families and
with them different types of stakeholders.
The decennial science outreach experience of the researchers involved in the activities highlighted in this paper was
characterized by a growing use of images and the use of games as educational tools, to raise students' awareness on
scientific issues. Researchers noted, in involved students, an appreciation of the use of this kind of approach as didactic
tool. This also emerges from the results of the questionnaire reported in the last paragraph, which shows that students
show more interest in educational-playful activities and information transmitted through images than in standard frontal
lectures.
It's easy to understand how through games and images it's easier to arise interest in younger learners; but this also applies
to all other ages, of course taking care to choose the right game and the right graphic style, depending on the target
audience. For these reasons, these methodological approaches are becoming increasingly useful in the field of scientific
dissemination.

**Acknowledgment**
The authors kindly thanks the artists and researchers that collaborated in the realization of the games: Matteo Sgherri and
Valentina Sgherri (OCTOPUS GAME), Consuelo Zatta (INGV-MEMORY), Francesca Di Laura and the INGV Graphic
Office (MEREOPOLI), Giacomo Guccinelli, Lucrezia Benvenuti, Mascha Stroobant and Roberta Talamoni (VISUAL
QUESTIONNAIRE). Moreover, a special thanks to the High Schools (Capellini-Sauro) that participated in PCTO and
the questionnaire, and to the researcher (Paolo Stefanelli) and Association (GAD - Gruppo Astronomia Digitale, in detail
Claudio Lopresti) that organized with us the activities of the PCTO. Authors thank the Historical Oceanography Society
too, for the support in the interpretation of historical topics then explained in MAREOPOLI.

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
