# Peer review of "Educational and artistic fun teaching tools for science outreach."

_Geoscience Communication, 2020_

## Referee Comment (RC1) · Anonymous Referee #1 · 21 Apr 2020

The presented work is interesting and touches important and new aspects about teaching tools. Few comments following:

a. the main touched areas are scientific dissemination and teaching. Are they different about targets and methods? Are artistic and visual elements differently intended and used in these two areas?

b. On my opinion the 'visual approach' is of basic importance, especially in teaching scientific matters. As a first stimulus, emotion induces the desire of knowing, and, one of the most privileged sense reaching our inside is the view. I think 1. the first act is the creation of right visual signs making concepts visually touchable, 2. the second act consists in making them artistically attractive, 1+2 results in 3, generating the 'emotion of learning'. Of course, the target sensitivity is a limiting factor: a 6 year

old child requires a different apporach vs. a 20 years old high school student. As an example, about the visual approach, Youtube is a source of many contributions. You can find many mathematical explanations; some are good in point 1, but scarce in point 2, and, viceversa. Some (actually, not many) are emotional in the sense of good 1+2=3 combination. Moving to text books, synonymous of more 'traditional teaching', it's harder to find a more efficient 1+2=3 combination. About teaching techniques, it should be nice to move towards a kind of 'emotion of learning' approach. On your perspection and experience, which are the limits to be overcome thinking about schools?

Please also note the supplement to this comment:
https://www.geosci-commun-discuss.net/gc-2020-6/gc-2020-6-RC1-supplement.pdf
* * *

---

## Author Comment (AC1) · 22 Apr 2020

Reviewer 1 thank you for your questions.

The answers have been elaborated on benhalf autor M. Locritani and co-author S. Merlino.

a) The teaching activity has to follow the scholastic curricula, the dissemination activity can extend the themes to other issues and increase the interest in science. The object of dissemination activities is to stimulate the emotions and interest in the scientific issues, with the aim to make more attractive the curricula matters. The images are important in both areas (teaching and dissemination), but in outreach activities the images always are the main vector of the information, differently in teaching activity

where the text is the main contribution.

b) In our opinion, the school curricola requires a too big courseloads knowledge and information, should be rethought and reduced to improve quality with respect to quantity. The practical and interactive approach, moreover, should be useful to explain the argument and interest students. The personal experience leaves an emotive track, that change the point of view of the students, increasing the interest and allowing to create a base to improve the learning of the future knowledge. In this plan the artistic and visive approach play a fundamental role.

---

## Referee Comment (RC2) · Anonymous Referee #2 · 21 May 2020

The paper presents an interesting approach on the use of game and illustration in the scientific outreach. The authors describe the realization of 4 different edutainment tools inspired from well know board games and based both on scientific content and visual impact. The game development process and the interaction between researchers and illustrators are well explained and motivated as well as the objectives, the range of users and the way the games are played. The introduction is too long and has parts that are repeated in the following paragraphs. The use of too long and complex sentences, with breaks in parentheses, makes reading in some cases difficult and unclear. An interesting analysis of questionnaires collected in connection with one game experience and with a group of users is presented. Considering that the developed games are 4 and that the activities have been carried out in 9 years, it would be interested if

the authors gave more feedback on the impact that these activities had on the users. Indications, even if only qualitative, of the level of satisfaction and impact on the improving in scientific knowledge. This is also valuable as in the introductory part of the paper the authors stressed the importance of a playful approach for better conveying scientific content.

It is not clear whether the VISUAL QUESTIONNAIRE has already been made, and therefore used and tested, or if it is still in progress. The training objectives are not clear either. The methods of interpretation of the results, which appear potentially interested, are also not adequately developed.

Specific comment and Technical corrections in the supplement file

Please also note the supplement to this comment:
https://www.geosci-commun-discuss.net/gc-2020-6/gc-2020-6-RC2-supplement.pdf

---

## Author Comment (AC2) · 23 Jun 2020

Dear reviewer,

I and the others authors are grateful for your comments and suggestions. We tried to eliminate superfluous phrases and shorten, where possible, in the introductory part. The on field experience of the authors in these nine years of outreach activity by using the playful approach are argumented from line 540 to line 543, moreover the utiliy of this approach was explained between line 139 to 146 in paragraph 2.2.1 Great succes of the laboratories, in detail of MAREOPOLI, make INGV invests on this issue, supporting the pubblication of an educational-game book: MAREOPOLI. The VISUAL QUESTIONNAIRE is a work in progress. We developed it starting from the "standard

paper questionnaire" for older students, in order to have the possibility to submit it also to pre-school children. It was conceived following the 5 bloks scheme of the standard questionnaire, but it needs a particular interpretation. The method of interpretation will be developed with the collaboration of psychologists and sociologists, and is currently being studied. In order to respond to the reviewer 's requests, we have added some parts (from line 440 to line 448), but we cannot currently devote too much space to the description of this methodology, either because it is not yet completed or because it is outside the objectives of this work. Followed below the answer to the specific comments.

Best regards,

Marina Locritani

Please also note the supplement to this comment:
https://gc.copernicus.org/preprints/gc-2020-6/gc-2020-6-AC2-supplement.pdf

**Supplement:**

Dear reviewer,

I and the others authors are grateful for your comments and suggetions.
We tried to eliminate superfluous phrases and shorten, where possible, in the introductory part.
The on field experience of the authors in these nine years of outreach activity by using the playful approach are argumented from line 540 to line 543, moreover the utiliy of this approach was explained between line 139 to 146 in paragraph 2.2.1 Great succes of the laboratories, in detail of MAREOPOLI, make INGV invests on this issue, supporting the pubblication of an educational-game book: MAREOPOLI.
The VISUAL QUESTIONNAIRE is a work in progress. We developed it starting from the "standard paper questionnaire" for older students, in order to have the possibility to submit it also to pre-school children. It was conceived following the 5 bloks scheme of the standard questionnaire, but it needs a particular interpretation. The method of interpretation will be developed with the collaboration of psychologists and sociologists, and is currently being studied. In order to respond to the reviewer 's requests, we have added some parts (from line 440 to line 448), but we cannot currently devote too much space to the description of this methodology, either because it is not yet completed or because it is outside the objectives of this work.
Followed below the answer to the specific comments.

Best regards,

Marina Locritani

Specific comments gc-2020-6

17-18 Use the legitimate original names of the Institutions – I did it.
replace *didactic* with educational – I did it.
remove *some* – I did it.
the year is missing in the reference – I added it.
*In our opinion* better at the beginning of the sentence – I corrected this.
36-37 rewrite the sentence in a more legible way – I corrected with: However, in our opinion, the "physicality" of the experience is important, and, even more in a technological era, it is necessary to invest resources to create stimuli that involve students in real activities.
39-40 6-14 years old students – I did it.
to make easier the text reading, put citations at the end of the sentence. if you cited in this way, the reader expects a citation in the references (that is not present) - I added it.
56- 58 – 122 -132- 145 to make easier the text reading, put citations at the end of the sentence – I did it.
to make easier the text reading, put citations at the end of the sentence. – I did it.
The citation is missing in the references – I added it.
94-96 rewrite the sentence in a more legible way – I rewrote: In particular, the large part of European (79%) and Italian (69%) people, are very interested and confident in new scientific discoveries and technological developments, however there is a part of respondents who do not feel as well informed (52% Italians, 50% Europeans).
the year is missing in the reference – I added it.
I didn't catch the relation between this example and the previous concept - I cut the example
replace *but* with However – I did it.
replace *didactic* with educational – I did it
176-177 remove parenthesis – I did it
for high school students – I did it
185-186- 187 remove the sentence after young children – I did it

195-196 remove the sentence – I did it to 206 remove parenthesis and *by students* – I did it workshops participants – I did it remove parenthesis and *but* – I did it remove parenthesis and *parties involved* – I did it replace *but it is precisely here that* with . In this contest – I did it remove parenthesis – I did it remove sentence in parenthesis – I did it citation of the figure at the end of the sentence – I did it

241 how? - thanks to the experience of the involved illustrators and researchers in this field the sentence it not clear – I rewrote: The spaces have 3 possible colours: blue, green, and yellow, each one corresponding to a specific topic, intuitively linked to the respective colour: water column, life in the sea and coast and seabed.

remove *some* – I did it remove parenthesis – I did it remove *that is* – I did it citation of the figure at the end of the sentence – I did it

265- 267 to do so the conductor should be very skilled on the topics and prepared on educational approach in contrast with what you say at line 268 – I dedde: , properly trained

- 274 too long sentence, not clear the meaning – I rewrote: The peer education has proved to be an efficient method to stimulate learning. The conductor students experiment an increase in self-esteem and self-confidence, which in turn conveys a greater retention of the concepts acquired during the experience. Moreover, the verbal and not verbal language of the conductor of the game is similar of that of participant, consequently more attractive how and for whom it may be available? – I added: A game-kit is available by the INGVambiente website (https://ingvambiente.com/2020/01/17/il-gioco-del-polpo/).

48 40x40 cm cards – I did it replace *covered* with game – I did it

Questions are already established in the game or is based on conductor knowledge? – I added: established by the game

301-302 not clear – I rewrote: In this case, however, what is represented in the different boxes has to do with the proposed topic, and sometimes it is also useful to suggest the answers of the questions.

which case? – I cut in this case citation of the figure at the end of the sentence – I did it

INGV Laboratorio di Grafica e Immagini (in italian) – I did it replace *Second Grade secondary* with High. Insert (Figure 7). After adults – I did it

337- 339 remove the sentence after adults – I did it

342- 352 – 504 INGV Laboratorio di Grafica e Immagini (in italian) – I did it replace *eras* with age – I did it primary and kindergarten school students – I did it

364-365 remove after students… to age – I did it remove sentence in the parenthesis – I did it

The story's arguments you use are critical for children's visual questionnaire interpretations. How do you handle it? I added these sentences: The 5 blocks of the graphic questionnaire are the same of the "standard" paper questionnaire with multiple choice answers previously used (Ref Locritani et al). We started from this questionnaire with the idea of extending it also to pre-school children. The methods of interpretation are currently still under development, with the help of psychologists and sociologists, and are based on the frequency with which some characters are used, depending on the "chosen kingdom", and which gadgets and dressing are used, as well as other parameters such as, for example, the preference of the "gender" of the character. These are, however, evaluations that will have to be chosen, among several possible, after having analysed the first "tests" performed on a small number of children, as we are currently trying to do through collaboration with some kindergartens and early elementary school. This part of the work is certainly very demanding and delicate, and would require a special treatment that goes beyond the scope of this exhibition.

This sentence it is not clear. The figure citation should be put at the end – I rewrote: and the citizens living in these 5 kingdoms (Figure 11) for a total of 67 characters (including avatar)  that will be photographed and analysed by researchers (Figure 11).

410- 411 Difficult to manage the analysis, in drawing children are free to represent and imagine, in this case the choice is necessarily limited to what you propose. it is not clear the block identification assessment…. – I added: The questionnaire has only administered to a small group of children for a test section. Interpretation of the questionnaire results will require collaboration with psychologists and sociologists. This part will be developed soon.

this is a critical point. Before stating that it will allow… you should test it – I don't understand

421-422 remove parenthesis and *and* – I did it what do you mean for unconventional educators? – I cut it

423-424 replace *Guccinelli….Factory)* with the Steam-Factory Team – I did it

423- 425 too long and not clear sentence – I rewrote: In this phase, collaboration with an expert designer, such as Steam-Factory Team, was fundamental, in order to create the appropriate framework for the questionnaire addressed to very young generations.

how do you probed? I added these sentences: Some preliminary tests were done in some primary (first three years) and kindergarden classes. During these tests, the children involved were asked to make drawings that concerned science and scientists. From this they drew some considerations emerged, regarding the use of the main characterizations preferred by children in this age group.

remove *regulate*…. 2019) – I did it remove *planetary* – I did it remove parenthesis – I did it

447- 454 figures citation at the end of the sentence – I did it remove *during PCTO experiences* – I did it remove figure citation and – I did it in parenthesis replace with figure – I did it not clear – I cut (as in this case)

replace *the concept( …)* with complex concepts – I did it

Add a conclusion paragraph expanding and arguing this final part. I added these sentences: The playful approach, that, in our opinion and for our experience is particularly useful for better conveying scientific content, lends itself very well to be combined with the use of images or other art forms. The synergy between these two "modes" is, in our opinion, an effective means to overcome potential barriers due, in many cases, to language, and promotes a greater diffusion of these tools.

Remove *We can assume that the* – I did it

Questionnaire result highlight the importance of… – I did it replace *indicate* with indicating – I did it

489-490 not clear – I rewrote: In addition, through the young generation is simple to reach their families and with them different types of stakeholders.

replace *didactic* with educational – I did it remove *We easy understand how* – I did it

Through games and image we can more easily arise… – I did it
replace *other ages* with users remove *of course* – I did it
replace *to choose* with in choosing – I did it
Francesca Di Laura is a co-author of the paper, so other acknowledgement are not necessary – I rewrote: Daniela Riposati (Laboratorio di Grafica e ImmagineINGV, Graphic Office)
-509 replace with
Moreover, a special thanks to the Capellini-Sauro High School that participated in PCTO, and to Paolo Stefanelli (INGV) and Claudio Lopresti (GAD - Gruppo Astronomia Digitale, that organized with us the PCTO activities. Finally, authors thank the Historical Oceanography Society for the support in the interpretation of historical topics then explained in MAREOPOLI – I did it

---

## Editor Comment (EC1) · Isaac Kerlow (Editor) · 24 Jun 2020

Thank you author(s) for posting your responses to the comments by the two reviewers.

Dear Anonymous Reviewers, please let us know if you have further comments.

Thank you and regards.

---

## Editor Decision (ED1)

Specific comments gc-2020-6

17-18 Use the legitimate original names of the Institutions replace *didactic* with educational remove *some*

the year is missing in the reference

*In our opinion* better at the beginning of the sentence

36-37 rewrite the sentence in a more legible way

39-40 6-14 years old students to make easier the text reading, put citations at the end of the sentence. if you cited in this way, the reader expects a citation in the references (that is not present)

56- 58 – 122 -132- 145 to make easier the text reading, put citations at the end of the sentence to make easier the text reading, put citations at the end of the sentence.
The citation is missing in the references

94-96 rewrite the sentence in a more legible way the year is missing in the reference

I didn't catch the relation between this example and the previous concept replace *but* with However, replace *didactic* with educational
176-177 remove parenthesis for high school students

185-186- 187 remove the sentence after young children

195-196 remove the sentence to 206 remove parenthesis and *by students*

workshops participants remove parenthesis and *but*

remove parenthesis and *parties involved*

replace *but it is precisely here that* with . In this contest remove parenthesis remove sentence in parenthesis citation of the figure at the end of the sentence

241 how?

the sentence it not clear remove *some*

remove parenthesis remove *that is*

citation of the figure at the end of the sentence

265- 267 to do so the conductor should be very skilled on the topics and prepared on educational approach in contrast with what you say at line 268

- 274 too long sentence, not clear the meaning how and for whom it may be available?

48 40x40 cm cards replace *covered* with game

Questions are already established in the game or is based on conductor knowledge?

301-302 not clear which case?

citation of the figure at the end of the sentence

INGV Laboratorio di Grafica e Immagini (in italian)

replace *Second Grade secondary* with High. Insert (Figure 7). After adults

337- 339 remove the sentence after adults

342- 352 – 504 INGV Laboratorio di Grafica e Immagini (in italian)

replace *eras* with age primary and kindergarten school students

364-365 remove after students… to age remove sentence in the parenthesis

The story's arguments you use are critical for children's visual questionnaire interpretations. How do you handle it?

This sentence it is not clear. The figure citation should be put at the end

410- 411 Difficult to manage the analysis, in drawing  children are free to represent and imagine, in this case the choice is necessarily limited to what you propose. it is not clear the block identification assessment….

this is a critical point. Before stating that it will allow… you should test it

421-422 remove parenthesis and *and*

what do you mean for unconventional educators?

423-424 replace *Guccinelli….Factory)*  with the Steam-Factory Team

423- 425 too long and not clear sentence how  do you probed? ..

remove *regulate*…. 2019)

remove *planetary*

remove parenthesis

447- 454 figures citation at the end of the sentence remove *during PCTO experiences*

remove figure citation and in parenthesis replace with figure not clear replace *the concept( …)* with complex concepts

Add a conclusion paragraph  expanding and arguing this final part.
Remove *We can assume that the*
Questionnaire result highlight the importance of…

replace *indicate* with indicating

489-490 not clear replace *didactic* with educational remove *We easy understand how*
Through games and image we can more easily arise…

replace *other ages* with users remove *of course*
replace *to choose* with in choosing

Francesca Di Laura is a co-author of the paper, so other acknowledgement are not necessary

-509 replace with

Moreover, a special thanks to the Capellini-Sauro High School that participated in PCTO, and to Paolo Stefanelli (INGV) and Claudio Lopresti (GAD - Gruppo Astronomia Digitale, that organized with us the PCTO activities. Finally, authors thank the Historical Oceanography Society for the support in the interpretation of historical topics then explained in MAREOPOLI